systems biology/bioinformatics/computational biology

periodic rhythms, cyclic attractors, state transition, optimal control

**Authors for correspondence:**
Weirong Hong
e-mail: hongwr@zju.edu.cn
Pu Li
e-mail: pu.li@tu-ilmenau.de

# Reconciling periodic rhythms of large-scale biological networks by optimal control

Meichen Yuan[1,2], Junlin Qu[2], Weirong Hong[1] and Pu Li[2]

[1]College of Energy Engineering, Zhejiang University, Hangzhou 310027, China
[2]Process Optimization Group, Institute of Automation and Systems Engineering, Technische Universität Ilmenau, Ilmenau 98684, Germany

MY, 0000-0002-6393-3858; WH, 0000-0001-8482-1225; PL, 0000-0001-6481-9961

Periodic rhythms are ubiquitous phenomena that illuminate the underlying mechanism of cyclic activities in biological systems, which can be represented by cyclic attractors of the related biological network. Disorders of periodic rhythms are detrimental to the natural behaviours of living organisms. Previous studies have shown that the state transition from one to another attractor can be accomplished by regulating external signals. However, most of these studies until now have mainly focused on point attractors while ignoring cyclic ones. The aim of this study is to investigate an approach for reconciling abnormal periodic rhythms, such as diminished circadian amplitude and phase delay, to the regular rhythms of complex biological networks. For this purpose, we formulate and solve a mixed-integer nonlinear dynamic optimization problem simultaneously to identify regulation variables and to determine optimal control strategies for state transition and adjustment of periodic rhythms. Numerical experiments are implemented in three examples including a chaotic system, a mammalian circadian rhythm system and a gastric cancer gene regulatory network. The results show that regulating a small number of biochemical molecules in the network is sufficient to successfully drive the system to the target cyclic attractor by implementing an optimal control strategy.

## 1. Introduction

Periodic rhythms are regular behaviours in biological systems [1]. For instance, the circadian clock helps regulate sleep schedule, body temperature, hormone levels in a *daily* cycle [2]; *monthly* rhythms are reflected in reproductive cycles of many marine plants and animals [3]; *annual* rhythms are expressed in flowering, migration, hibernation or the reproduction and growth of most terrestrial plants and animals in temperate zones [4]. If the natural biological rhythms are disturbed, disorders of the organisms may

arise, e.g. sleep disorders for human beings [5]. All these disorders may result in daytime sleepiness, depression, arrhythmogenesis and even cancer [6–9]. In general, the organism can find itself a way to adjust the disturbed rhythms to the regular ones in the new environment. But this recovery process usually takes too much time for the biological system to suffer from. Therefore, an external intervention can be used to assist the biological system to accelerate the recovery process. For this purpose, investigations on where and how to intervene in the biological system under consideration have to be made.

Multiple steady states of a biological network can be represented by attractors, generally in the form of point attractors, cyclic attractors, chaotic attractors and the like [10,11]. Multi-stability plays a crucial role in biochemical networks at many levels from genes, cells, tissues to organs, exhibiting different kinds of phenotypes or various functions [12,13]. Point attractors represent steady states of different cell types, for example, megakaryocytes, erythrocytes, granulocytes and monocytes proliferated and differentiated by a common myeloid progenitor in haematopoiesis [14]. Chaotic attractors are hypothesized to associate with patterns in odour recognition [15]. Cyclic attractors describe oscillatory behaviours like walking or chewing governed by neurons in animals and circadian rhythms caused by day and night changes. Therefore, periodic rhythms can be considered as cyclic attractors of the related biological system [1]. Based on the concept of attractors, a biological system is either working around an attractor or transferring from an attractor to another one, which can be regulated by some species or regulatory interaction strengths in the network [16]. Indeed, the past decade has witnessed extensive studies in the controllability of biological systems and shed light on the underlying mechanism of state transitions [17–20]. However, in most of these studies approaches were proposed to identify driver nodes aiming at state transitions between point attractors [21–26], while cyclic attractors have been ignored.

To hold the regular periodic rhythms of a biological system, it is necessary to keep the amplitudes and phases of the state variables at the desired cyclic attractor. More importantly, if the system is working at an abnormal attractor, a state transition between cyclic attractors or other types of attractors is required. It means that a measure of intervention is needed. For such a purpose, we need at first to identify some proper regulators (i.e. intervention or control variables) inside the network and then to determine control profiles for the intervention. Several approaches have been developed to control cyclic rhythms. Slaby *et al.* [27] and Shaik *et al.* [28] applied model-based optimal control to find appropriate strength and timing of light stimulus to suppress or restore the circadian rhythms of the Drosophila model. In these studies, light-sensitive parameters were taken as control variables *a priori*, i.e. they only determined the profiles of the control variables. Based on the graph theory, Fiedler *et al.* [29] and Mochizuki *et al.* [30] developed a method to identify a set of state variables in a network, a so-called feedback vertex set. The feedback vertex sets are prescribed and the remaining state variables are obtained by the model equations to follow the trajectories of a target attractor. However, the application of this method can be limited since directly forcing a set of state variables with prescribed trajectories may be unrealistic in clinical trials. Model predictive control was applied to manipulate the mammalian circadian clock by Abel & Doyle [31] for phase resetting. Moreover, Jin *et al.* [32] proposed a two-step solution strategy, i.e. identifying the driver nodes at first and then designing a closed-loop controller for steering the system towards the desired attractor.

Although these existing methods can help to optimally identify control variables and determine their profiles for state transition, no method is available to simultaneously optimize both, especially for the purpose of reconciling periodic rhythms of large-scale biological networks. Obviously, control variable identification and profile determination should be considered simultaneously, so that the coupling effect of both tasks involved in the optimization problem can be taken into account. In a recent study, we proposed an optimization approach to simultaneously identify regulatory variables and determine their profiles for state transition of biological networks [33]. A mixed-integer nonlinear dynamic programming (MINDP) problem is solved with the purpose to steer the system to a desired point attractor.

In the current study, we extend the method in [33] to investigate an optimal control approach for the reconciliation of periodic rhythms. To elaborate cyclic trajectories of state variables for a state transition, we define three time periods to describe the optimization process. The first time period portrays the state of the initial periodic rhythms, the second one relates to the regulation process and the third one is used to validate the qualification of the state trajectories at the target attractor. It means that we require the system to arrive at the desired cyclic attractor in the third time period. In our approach, the identification of the control variables is associated with the binary variables and the profiles of such decision variables are time-dependent intervention strategies. Both binary variables and the profiles of control variables will be optimized simultaneously by solving a MINDP problem. In addition, the regulation time which corresponds to the second time period will be determined based on the

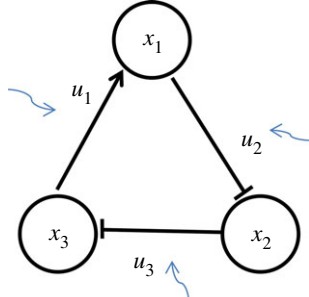

**Figure 1.** Schematic description using a simple system.

operating condition specified by intervention restrictions. Furthermore, our approach allows finding out a minimum number of control variables for the intervention. Three case studies including a chaotic system, a mammalian circadian system and a gastric cancer system are implemented to verify the effectiveness of the proposed approach.

# 2. Material and methods

## 2.1. Problem description

In general, the dynamics of a biological network can be expressed as a set of $N$-coupled ordinary differentiable equations (ODEs):

$$\frac{\mathrm{d}x_i}{\mathrm{d}t} = f_i(x_1, \ x_2 \ , \ \cdots, \ x_N, p) - k_i x_i, \ x_i \geq 0, \ i = 1, \ldots, N, \tag{2.1}$$

where $x_i$ is the $i$th state variable, $f_i$ is the continuously differentiable function describing the interaction between the state variables, $N$ is the number of biological components, $p$ is the parameter set. The state variables correspond to the concentrations of the components in the biological network which are mutually interacting, exhibiting activation or inhibition, with coupling interaction parameters. In equation (2.1), $k_i x_i$ is the self-degradation term of the biological component, where $k_i$ is the degradation rate.

According to Gardner *et al.* [34], the injection of drugs in the cell-growth environment assists in controlling the fate of the cells by adjusting the interaction parameters. In addition, from the work of Li & Wang [35], changing the strength of an activation or repression regulation parameter in the gene regulatory network helps the realization of a state transition. In such cases, it is assumed that these parameters can be regulated by a kind of external intervention through the application of repressive or inductive drugs as made in the work of Wang *et al.* [20]. Therefore, the strength of the interaction can be considered as a control variable. Figure 1 shows a simple system with three components, where the black arrow-head edges and the black bar-head edges mean the activation regulation and the inhibition regulation from the source node to target node, respectively; $u_1$, $u_2$ and $u_3$ are the interaction strengths from $x_3$ to $x_1$, from $x_1$ to $x_2$ and from $x_2$ to $x_3$, respectively. The blue curve arrows indicate that an intervention can be implemented by adjusting the values of $u_1$, $u_2$ or $u_3$ for the control of this system.

In this study, we take the average interaction strength of a component with the other components as a candidate for a control variable. As a result, we extend equation (2.1) to

$$\frac{\mathrm{d}x_i}{\mathrm{d}t} = u_i f_i(x_1, \ x_2 \ , \ \cdots, \ x_N, p) - k_i x_i, \ x_i \geq 0, \ i = 1, \ldots, N, \tag{2.2}$$

where $u_i$ represents the average interaction strength from the other components to the target component $i$. Equation (2.2) means that $u_i$ includes its own feedback regulation but does not contain the degradation reaction. Since only a few can be selected as control variables for an intervention, we need to decide which ones should be selected. Comparing with equation (2.1), it can be seen from that, if the $i$th component is selected as a control variable, then $u_i \neq 1.0$, i.e. an action will be performed to change the average interaction strength. If it is not selected, then $u_i = 1.0$, i.e. the average interaction strength remains its regular value. Also, the control variables defined in this way are time-dependent, since an intervention strategy in a time period is needed. In addition, in comparison to equation (2.1), the nominal value of all control variables is 1.0. Therefore, for performing an intervention, the initial

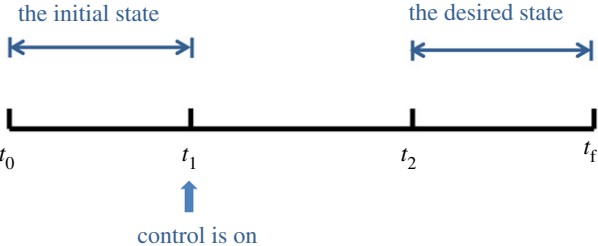

**Figure 2.** Time periods in the optimization process.

value of all the control variables is 1.0, and after the intervention, it should return to this value to ensure the stability of the system.

In this study, we use the method of optimal control to simultaneously select proper control variables and determine their control profiles. The objective function is defined as the minimization of the deviation from the desired attractor integrated over a period of intervention time, as follows:

$$\min \int [x(t) - x^{\text{des}}(t)]^2 \mathrm{d}t + \int \left[\frac{\mathrm{d}x(t)}{\mathrm{d}t} - \frac{\mathrm{d}x^{\text{des}}(t)}{\mathrm{d}t}\right]^2 \mathrm{d}t, \tag{2.3}$$

where $x(t)$ is the state vector corresponding to the abundance/activity of the biochemical components in the biological network, while $\mathrm{d}x(t)/\mathrm{d}t$ represents the time derivative of the state variables. $x^{\text{des}}(t)$ and $\mathrm{d}x^{\text{des}}(t)/\mathrm{d}t$ are the state and time derivative vector of the desired attractor, respectively. It is to note that the minimization of the deviation of the time derivatives has not been considered in previous studies on state transition. In this study, we introduce this term in the objective function, since dynamic behaviours corresponding to the desired periodic rhythms are to be followed.

According to equation (2.3), both the trajectories of the state variables $x^{\text{des}}(t)$ and their derivatives $\mathrm{d}x^{\text{des}}(t)/\mathrm{d}t$ at the desired attractor have to be known. To achieve such trajectories, we need to identify the attractors of the biological network. For this purpose, one can solve the model equations (i.e. equation (2.1)) with randomly sampled initial conditions [21,36]. With a large number of samplings and as time $t$ is long enough, the stable behaviours at the attractors can be identified and the trajectories of $x^{\text{des}}(t)$ and $\mathrm{d}x^{\text{des}}(t)/\mathrm{d}t$ obtained. In addition, the periodic lengths corresponding to the cyclic attractors are also determined.

## 2.2. Definition of time periods

Considering the complexity of state transition from an initial cyclic state into an attractor with desired periodic rhythms, we define the optimization process in three time periods, as illustrated in figure 2.

As shown in figure 2, the time horizon for the optimization is defined as $[t_0, t_f]$. The first time period $[t_0, t_1]$ is used to express the initial state where $t_0$ is the initial time point of the optimization process. The length of this time period should be higher than the periodic length of the initial rhythm $T_1$, so that the initial state of the system can be clearly recognized. The second time period $[t_1, t_2]$ is the epoch of the intervention $T$, i.e. the control variables will be activated in this time period. $[t_2, t_f]$ is the time period where the system is driven into the desired trajectories of the target state and the intervention is withdrawn. $t_f$ is the end time point of the whole optimization process. Therefore, the time period $[t_2, t_f]$ should cover at least one periodic length of the target attractor $T_2$. As a result, the whole time period $[t_0, t_f]$ can be specified as $t_f - t_0 \geq T_1 + T + T_2$. From the state transition point of view, the epoch of the intervention $T$ should be as short as possible. However, if it is specified too short, a much stronger action of the control variables will be required. In this study, we initialize a relatively large intervention epoch $T$ and define the integration time in equation (2.3) as $[t_1, t_f]$. Since the desired state will be followed in $[t_2, t_f]$ after the intervention, we can determine the optimized intervention epoch $T$ from the optimization approach.

## 2.3. Identification of control variables

As discussed in §2.1, among $N$ candidates in equation (2.2), we need to identify a set of control variables on which the intervention should be implemented. For this purpose, we introduce binary decision variables $y$. $y_j = 1$ means that the corresponding interaction strength $u_j$ will be chosen as a control

variable $u_j(t)$; $y_j = 0$ means that $u_j$ will be a constant, i.e. $u_{j0} = 1$. As a result, the relationship between $y_j$ and control $u_j$ is expressed as

$$\begin{cases} u_j(t) = u_j(t), & \text{if } y_j = 1 \\ u_j(t) = u_{j0}, & \text{if } y_j = 0. \end{cases} \tag{2.4}$$

To integrate this relationship in the optimal control problem, we reformulate equation (2.4) as

$$\begin{aligned} u_j(t) &\geq u_{j0} + (u_j(t)_{\min} - u_{j0}) \cdot y_j \\ u_j(t) &\leq u_{j0} + (u_j(t)_{\max} - u_{j0}) \cdot y_j. \end{aligned} \tag{2.5}$$

From equation (2.5), $y_j = 1$ leads to $u_j(t)_{\min} \leq u_j(t) \leq u_j(t)_{\max}$, and $u(t)_{\min}$ and $u(t)_{\max}$ are the boundaries for the strength of the control variables $u(t)$, while $y_j = 0$ leads to $u_j(t) = 1.0$.

In order to determine the total number of control variables to be selected, let $\sum y_j = N_{\text{CON}}$, i.e. $N_{\text{CON}}$ is the sum of the binary variables which is a prescribed number. At the beginning, one can define $N_{\text{CON}} = 1$ in the problem formulation. If the resulting solution can drive the system to the desired attractor, it means that one intervention variable is enough to realize the state transition. Otherwise, $N_{\text{CON}}$ needs to be increased by one with which the optimal control problem will be solve once again. This procedure proceeds until the target attractor is achieved and allows determining a minimum number of control variables for the intervention. To check if the system is driven into the desired attractor, the time period of one cycle of the target attractor before time point $t_f$ is taken. Inside this periodic length, the value of the integrated deviation in equation (2.3) will be evaluated and should be less than a predefined threshold.

## 2.4. Formulation of the optimal control problem

Based on the above analysis, to simultaneously identify a set of optimal control variables and determine their profile, we define the following mixed-integer nonlinear dynamic programming (MINLDP) problem:

$$\min \sum_{i=1}^{N} \left( \int_{t_1}^{t_f} \alpha_i \cdot [x_i(t) - x_i^{\text{des}}(t)]^2 dt \right) + \sum_{i=1}^{N} \left( \int_{t_1}^{t_f} \beta_i \cdot \left[ \frac{dx_i(t)}{dt} - \frac{dx_i^{\text{des}}(t)}{dt} \right]^2 dt \right) + \sum_{j=1}^{M} \left( \int_{t_1}^{t_f} \gamma_j \cdot [u_j(t) - u_{j0}]^2 dt \right)$$

subject to equation (2.2)

equation (2.5)

$$\sum_{j=1}^{M} y_j = N_{\text{CON}}$$

$$y \in \{0,1\}$$

$$x(t) = x^{\text{init}}(t), t_0 \leq t \leq t_1$$

$$x(t)_{\min} \leq x(t) \leq x(t)_{\max}$$

$$u(t)_{\min} \leq u(t) \leq u(t)_{\max}$$

$$t_0 \leq t \leq t_f, \tag{2.6}$$

where $N$ is the number of state variables or biochemical components, $M$ is the total number of regulatory factors among which $N_{\text{CON}}$ will be identified as control variables, $u_0 = 1.0$ is the initial average interaction strength, and $x^{\text{init}}$ is the initial state for the biological networks, respectively. $x(t)_{\min}$ and $x(t)_{\max}$ are the boundary values of the abundance or activity of biochemical components $x(t)$.

In the objective function, the aim of the first two terms $\sum_{i=1}^{N} \left( \int_{t_1}^{t_f} \alpha_i \cdot [x_i(t) - x_i^{\text{des}}(t)]^2 dt \right)$ and $\sum_{i=1}^{N} \left( \int_{t_1}^{t_f} \beta_i \cdot [(dx_i(t)/dt) - (dx_i^{\text{des}}(t)/dt)]^2 dt \right)$ is to make sure that the desired trajectories of the target state be approached, i.e. the deviation between the target and realized trajectories should be minimized in the time period $[t_1, t_f]$. It is noted that there exist bifurcation points in a multi-stable system [20]. Therefore, once the intervention is large enough to steer the system to go through bifurcation points, the initial steady state will lose its stability. As a result, the system will be steered to the desired state [15]. The third term in the objective function $\sum_{j=1}^{M} \left( \int_{t_1}^{t_f} \gamma_j \cdot [u_j(t) - u_{j0}]^2 dt \right)$ avoids excessive control actions so as to ensure physically meaningful profiles of the intervention variables. $\alpha$, $\beta$ and $\gamma$ are weighting factors which can be tuned to obtained expected optimization results. $\alpha$ and $\beta$

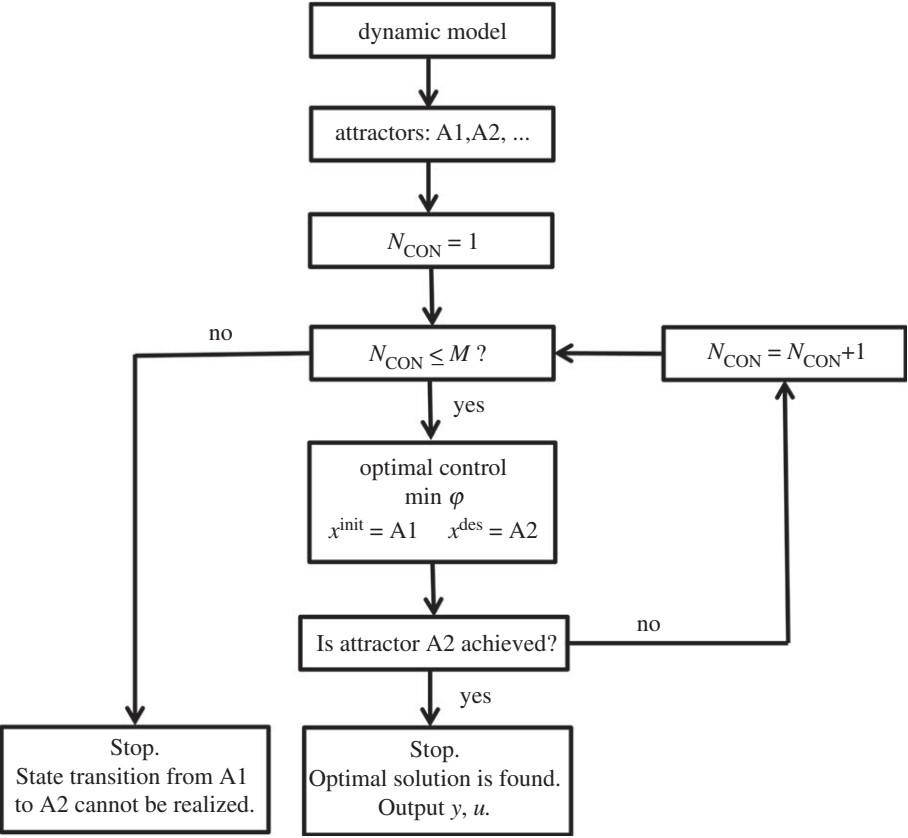

**Figure 3.** The flowchart for state transition of biological networks by optimal control.

can be tuned to make sure to follow the trajectory of the desired steady state, while $\gamma$ is associated with the dosage strategy of the drug. The selection of values for different parameters including the weighting factors in our work is listed in electronic supplementary material, data 1.

There have been several methods of solving MINLDP problem. Without loss of generality, we discretize the dynamic system with ODEs into a set of algebraic equations by the collocation method [37,38]. After the discretization, the MINLDP problem is transformed into a mixed-integer nonlinear programming (MINLP) problem. In this study we use the solver SBB [39] in GAMS [40] to solve this problem. GAMS is known as a high-level modelling system for mathematical programming and optimization. SBB combines the standard branch and bound method with some of the NLP solvers, which have been proved to be able to solve many problems very effectively [41]. To better understand the proposed optimization approach, the GAMS codes of the three case studies from different models are provided in electronic supplementary material.

The whole algorithm for reconciling periodic rhythms of biological networks by optimal control is illustrated in figure 3. It is assumed that the biological network has different attractors A1, A2, A3 and so on. Herein the transition from A1 to A2 is taken as an example to explain the process. As mentioned in §2.3, $N_{CON}$ needs to be increased one by one until the optimal solution is found or the state transition is judged to fail.

# 3. Results

## 3.1. A chaotic system

We first take a chaotic system [42] as a simple example to demonstrate the effectiveness of our method. The system is described as the following ODEs:

$$\left.\begin{array}{l} \dot{x}_1 = u_1 * x_2 * x_3 + 0.01, \\ \dot{x}_2 = u_2 * (x_1^2 - x_2) \\ \dot{x}_3 = 1 - 4 * u_3 * x_1, \end{array}\right\} \tag{3.1}$$

and

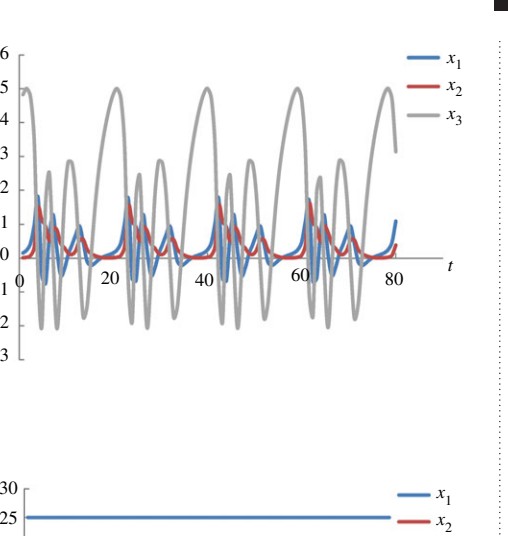

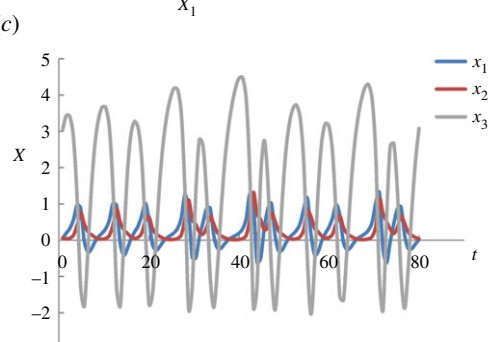

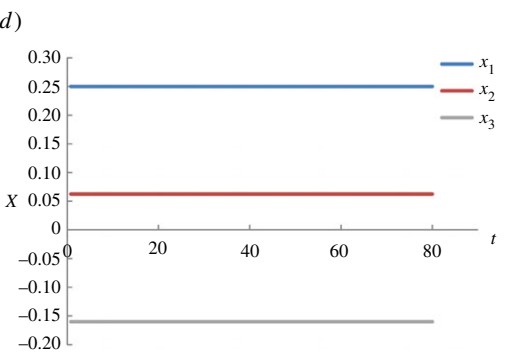

**Figure 4.** Attractors of the chaotic system. (*a*) Three-dimensional phase plane showing the three attractors: black bold line is the cyclic attractor (CYA); the red line is the chaotic attractor (CHA); the blue point is the point attractor (POA). (*b*) Trajectories of state variables in CYA. (*c*) Trajectories of state variables in CHA. (*d*) Trajectories of state variables in POA = [0.25, 0.0625, −0.16].

where $u_1$, $u_2$ and $u_3$ are the average regulatory interaction strength, and $x_1$, $x_2$ and $x_3$ are the states of the biochemical components, respectively. The initial values of $u_1$, $u_2$ and $u_3$ are defined as 1.0. This system has three different types of attractors, one cyclic attractor, one chaotic attractor and one point attractor, denoted as CYA, CHA and POA, respectively. These attractors are shown in figure 4. As mentioned in §2.2, the time horizon for the optimization [$t_0$, $t_f$] should be defined as longer than the sum of the periodic length of the initial attractor, the periodic length of the target attractor and the epoch of the intervention. The periodic length of CYA is 19.2, and thus we define $t_0 = 0$, $t_f = 80$. The state transitions between POA, CYA and CHA are accomplished using our optimization approach. The resulting optimal transitions from CHA to POA and from CHA to CYA are shown in figure 5.

The transition from CHA to POA is illustrated in figure 5*a*. The initial trajectories of the state variables at CHA are shown in the time period [0, 32] in the top of figure 5*a*. During this time period, there is no intervention and hence the values of the control variable remains at 1.0, as seen in the middle of figure 5*a*. From time $t_1 = 32$ the control variable $u_3$ is active and its profile is determined by the optimization approach. After a time period of intervention between time $t_1 = 32$ and time $t_2 = 46.4$, the system is driven into the attractor POA as shown in the time period [46.4, 80], in which the control variable $u_3$ restores to its original value, i.e. $u_3 = 1.0$.

Figure 5*b* shows the state transition from CHA to CYA. The initial state CHA is shown by the solid line in the first time period [0, 32] in the top. In this scenario, when the control is on from time $t_1 = 32$, the intervention determined by the optimization method is performed by manipulating $u_1$. The epoch of intervention is between time $t_1 = 32$ and time $t_2 = 42.4$. Then, the system is driven into the cyclic attractor CYA. It can be seen from the top of figure 5*b*, in the time period [42.4, 80] the trajectories of state variables coincide with those of the desired cyclic attractor denoted by the dashed lines. Again, the control variable $u_1$ restores to its original value after the intervention. At the solution, the integral value of equation (2.3) in the time period [60.8, 80] (i.e. one periodic length of CYA before $t_f$) is 0.0064, which is less than the predefined threshold 0.01. It means that the initial state CHA is indeed steered into the desired state CYA.

From this example, it can be seen that, although a long time horizon (i.e. 80) is defined for the optimization, the resulting intervention time is short, e.g. 14.4 and 10.4 for the two cases shown in

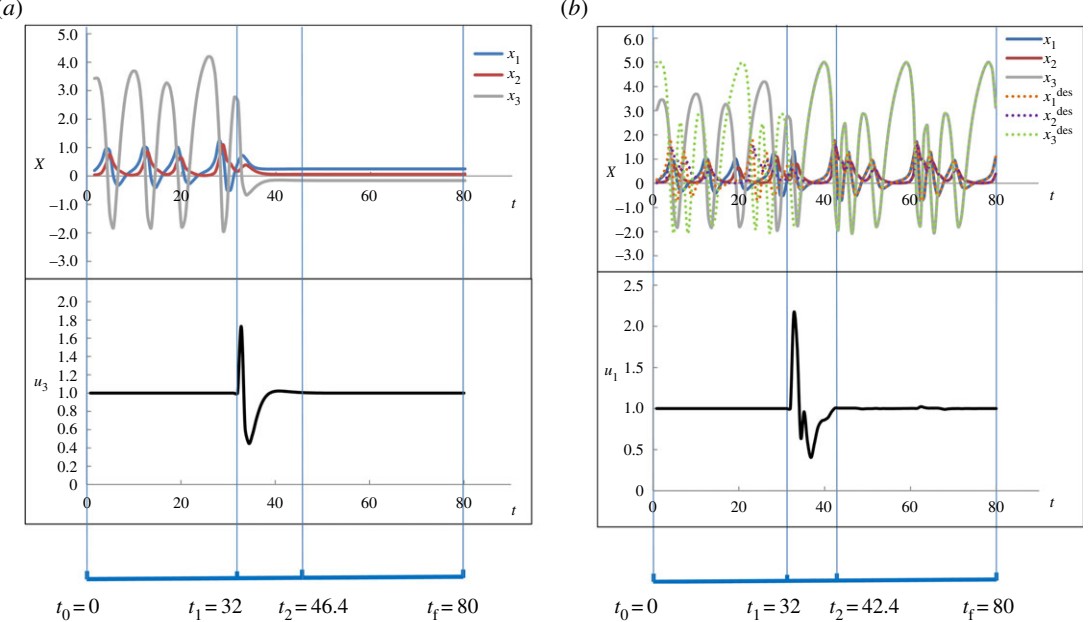

**Figure 5.** The optimization results for state transitions in the chaotic system. The top indicates time tracks of state variables; the middle is the profile of control variables $u_3$; the bottom shows the time periods. (a) The transition from CHA to POA, (b) The transition from CHA to CYA.

figure 5, respectively. It means that the results will not be dependent on the length of the time horizon, if it is defined long enough. It should be noted that among the values of $t_0$, $t_1$, $t_2$ and $t_f$, only $t_2$ is determined by the optimization approach.

In addition to the above two cases shown, the optimal control strategies for the transition from CYA to CHA, from CYA to POA, from POA to CYA and from POA to CHA are illustrated in electronic supplementary material, data 2. In all these cases, our method is able to identify the control variables and their profiles simultaneously. It can be seen that one single control variable is enough to realize the state transitions in the chaotic system. Taking the state transition from CHA to POA as an example, to study the effect of different parameter sets on the results of the proposed method, we took ten sets of different guess values of state variables and control variables (see electronic supplementary material, data 2) in the optimization approach. The results obtained are the same as the one shown in figure 5a, which means that our approach for identifying the control variables is robust.

## 3.2. Mammalian circadian rhythms

Circadian rhythm is a very common phenomenon in the biological activities, exhibiting an endogenous 24 h oscillation in behaviour, physiology and metabolism [43]. It plays an important role in a lot of biological processes and physiological functions including mammalian blood pressure and body temperature, sleep/wake cycles, the movement of leaves and the opening of flowers, and the like [44]. Disruptions of a normal circadian clock will result in metabolic dysregulation which is related to disease behaviours [45,46]. In that case, the identification of regulatory variables and performing an intervention to the circadian system so that it returns to the regular state are necessary. Mathematical models of circadian rhythms have been developed for understanding of the underlying mechanism and designing the manipulation of the system *in silico* [47].

Here, a mathematical model proposed by Mirsky *et al.* [44] with the parameter values from the work of Mochizuki *et al.* [30] is chosen. The model of biological network has 21 state variables and 132 parameters (see electronic supplementary material, data 3). In particular, we consider the phase delay induced by jet lag or shift work which is to be compensated by state transition using our optimal control method. At first, four attractors are identified by simulation, including two stable cyclic attractors (CYA1 and CYA2), one unstable cyclic attractor (CYA) and one unstable point attractor (POA), which is consistent with the finding of Mochizuki *et al.* [30]. For the unstable attractors, they may remain stable for a while, but when the time is long enough, they will eventually evolve into a stable attractor. Figure 6

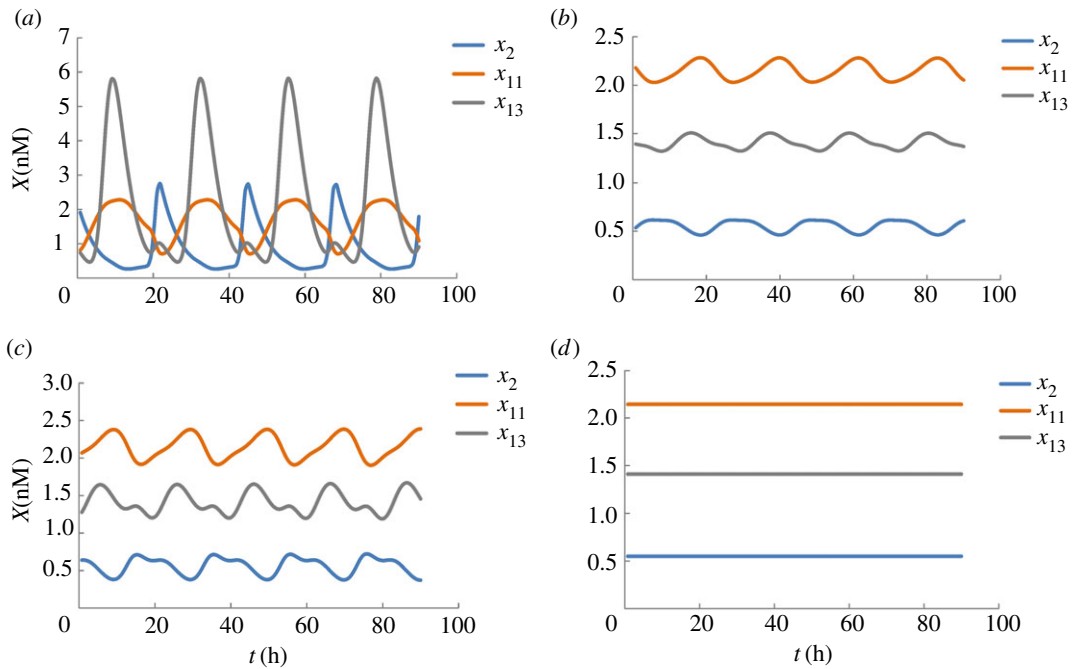

**Figure 6.** The trajectories of 3 state variables of mammalian circadian rhythms at different attractors. (*a*) CYA1. (*b*) CYA2. (*c*) CYA. (*d*) POA.

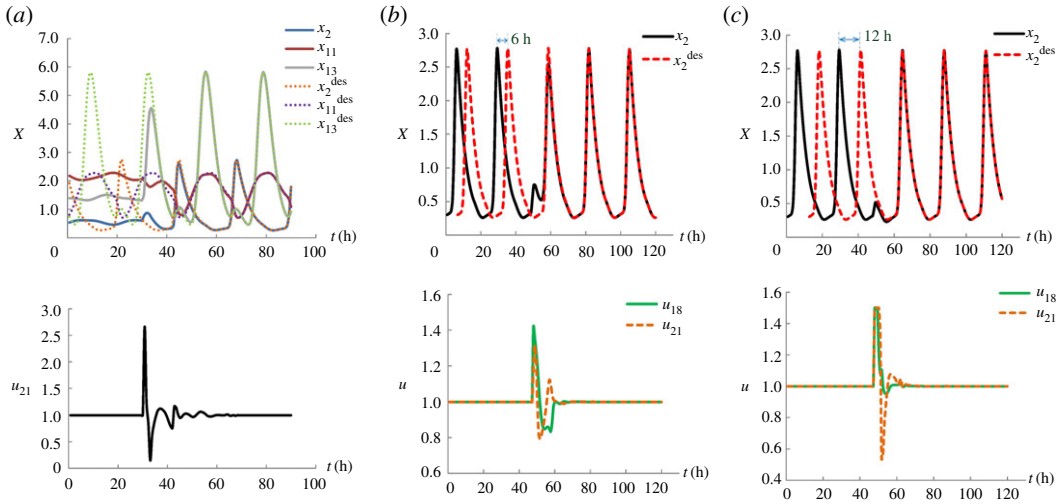

**Figure 7.** Optimization results for mammalian circadian rhythms. (*a*) State transition from CYA2 to CYA1. (*b*) Phase resetting (6 h) strategy caused by a jet lag. (*c*) Phase-resetting (12 h) due to shift work.

shows the trajectories of three state variables ($x_2 = Per2$, $x_{11} = CRY1$, $x_{13} = REV - ERB\alpha$) to illustrate the four different attractors. In addition, the periodic length of CYA1, CYA2 and CYA is 23.25, 21.75 and 20.25 h, respectively.

According to Abel & Doyle [31], high amplitude of biochemical components is preferable for enhanced metabolic health. Diminished circadian amplitude due to diet or age in mice is found to cause diseases including metabolic syndrome and diabetes [48–50]. In addition, sleep disorders from jet lag or shift work relate to abnormal cell growth and cancer formation [5,51]. In this example, we study the state transition from CYA2 to CYA1 to illustrate the acquisition of high-amplitude metabolic behaviour and choose CYA1 to demonstrate phase resetting to ease the effects of shifting circadian timing or amplitude after jet lag or shift work. Results are shown in figure 7.

As shown on the top of figure 7*a*, the dashed lines represent the trajectories with high-amplitude at the desired attractor CYA1, and the states in solid lines with initial low-amplitude at CYA2 are plotted in the first time period [0, 30 h]. After the optimal control, the trajectories of the state variables (solid lines)

**Table 1.** Impact of different weighting factors on state transition from CYA2 to CYA1.

| $\alpha = \beta$ | $\gamma$ | state transition | $U^L$ | $U^U$ | epoch of intervention |
|---|---|---|---|---|---|
| 1 | 1 | success | 0.149 | 2.662 | [30 h, 57 h] |
| 1 | 2 | success | 0.311 | 2.662 | [30 h, 57 h] |
| 1 | 4 | success | 0.472 | 2.662 | [30 h, 57 h] |
| 1 | 10 | success | 0.678 | 2.414 | [30 h, 57 h] |
| 1 | 50 | success | 0.900 | 1.782 | [30 h, 57 h] |
| 1 | 100 | success | 0.940 | 1.549 | [30 h, 57 h] |
| 1 | 200 | success | 0.929 | 1.361 | [30 h, 57 h] |
| 1 | 500 | fail | — | — | — |

are consistent with the desired CYA1 (dashed lines), demonstrating the realization of the state transition. The control variable $u_{21}$ (i.e. the average regulatory interaction strength from the other components to CLK/BMAL1) is identified and the profile of $u_{21}$, as shown at the bottom of figure 7a, is determined.

The recovery from a phase delay is shown in figure 7b,c. The trajectory of the state variable $x_2$ in attractor CYA1 is shown to illuminate the optimization results. The scenario in figure 7b associates with a 6 h time difference due to a flight from China to Germany and that in figure 7c corresponds to a shift work with 12 h. On the top plot of figure 7b,c, the black solid lines in the first time period [0, 47.25 h] represent the trajectory of original biological clock and the red dashed lines are the desired trajectory in the new environment. It can be seen that, after the optimal control, the dynamic system is successfully adjusted to adapt to the new environment. In these two scenarios, two control variables $u_{18}$ (the average regulatory interaction strength from the other components to PER2/CRY1) and $u_{21}$ are identified and their corresponding profiles are obtained, as shown in the bottom plot of figure 7b and c, respectively. This result reflects the important role of PER2/CRY1 and CLK/BMAL1 in the mammalian circadian rhythms as illustrated in the work of Mirsky et al. [44].

To study the impact of the weighting factors on the optimization results, several numerical experiments with different $\alpha$, $\beta$ and $\gamma$ values for the state transition from CYA2 to stable CYA1 are performed and the results are shown in table 1. In the objective function in equation (2.6), the first two terms evaluate the deviation between the trajectories of state variables and those of the target state variables, while the third term assesses the deviation of the control variables from their initial value. Therefore, we increase the value of the weighting factor $\gamma$ to investigate its impact on the control profile for the intervention. Also, considering the same physical meaning, $\alpha$ is prescribed to be equal to $\beta$. In table 1, $U^U$ and $U^L$ are the maximum and minimum value of the profile of $U21$ resulted by our optimization, respectively. It can be seen that, as the $\gamma$ value increases, $U^L$ is increasing and $U^U$ is decreasing, i.e. the magnitude of change of $U21$ is reduced for the intervention. However, if $\gamma$ is too large, the state transition will fail ($\gamma = 500$ in this case study), it means that the control action is too weak to trigger the state transition. In addition, different combinations of weighting factors have little impact on the epoch of intervention. Therefore, weighting factors mainly influence the control profiles of the identified decision variables.

## 3.3. Gastric cancer gene regulatory network

Not only are the periodic rhythms present within individual cells and at the tissue and organismal levels, but also they are common from the aspect of genetic level and even associated with cancer development. Here, we use the gastric cancer gene regulatory network constructed by Li et al. [52] to demonstrate the adjustment of periodic rhythms by optimal control. This molecular network has a relatively complex structure with 48 components and 215 regulation edges as shown in figure 8a. The model of the dynamic system with 48 ODEs is given in electronic supplementary material, data 3. By simulation, seven attractors are figured out, including six point attractors and one cyclic attractor (CYA) as shown in figures 8b,c. Among the seven attractors, POA1, POA2 and POA3 are known as cell cycle arrest of the normal cells, POA4 and POA5 are related to stress response, POA6 is regarded as the proliferation of cancer cells and cyclic attractor CYA indicates cell death. The detailed values of these attractors are listed in electronic supplementary material, data 4. These specific cellular phenotypes found by our simulation study coincide with the previous findings by Li et al. [52].

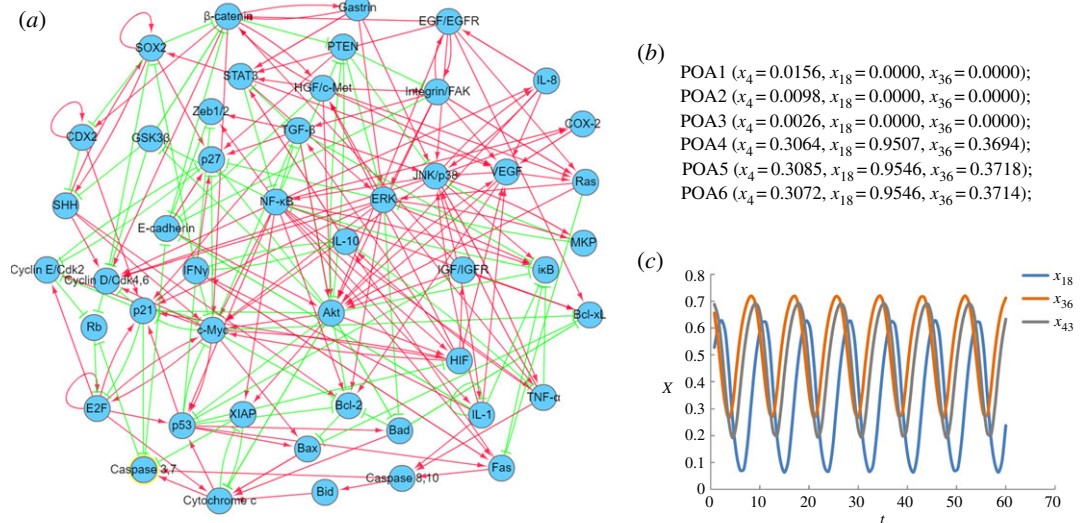

**Figure 8.** Gastric cancer gene regulatory network and attractors. (*a*) The molecular network of gastric cancer redrawn from Li *et al*. [52] by Omicshare (http://www.omicshare.com/tools), in which the nodes represent different kinds of components, the red arrow-head edges and the green bar-head edges mean the activation regulation and the inhibition regulation, respectively. (*b*) Six point attractors represented by the activity of three components $x_4$, $x_{18}$ and $x_{36}$. (*c*) The cyclic attractor (CYA) represented by $x_{18}$, $x_{36}$ and $x_{43}$.

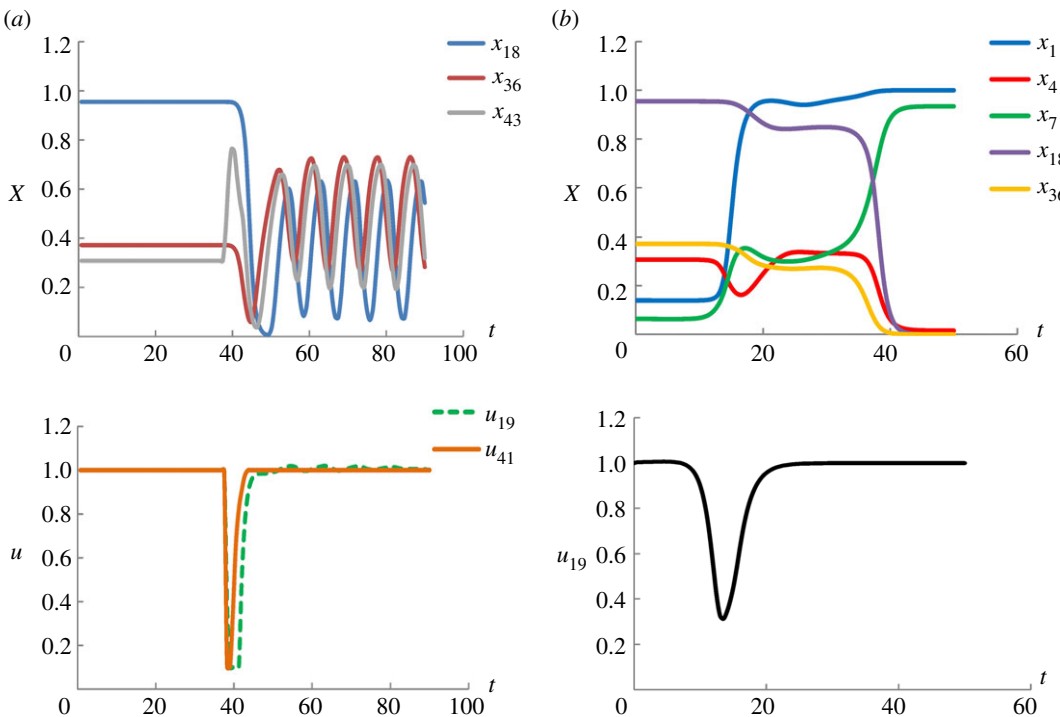

**Figure 9.** Optimal control for gastric cancer network. (*a*) The state transition from gastric cancer attractor (POA6) to apoptosis attractor (CYA). (*b*) The transition from gastric cancer attractor (POA6) to normal gastric epithelial attractor (POA1).

The aim of this example is to illustrate the efficacy of optimal control to prevent the cancer occurrence or transform the initial cancerous state back to the normal state. Thus, we study the transition from the gastric cancer attractor (POA6) to the apoptosis attractor (CYA) and from the gastric cancer attractor (POA6) to the normal gastric epithelial attractor (POA1) and the results are shown in figure 9. In figure 9*a*, three state variables ($x_{18}$, $x_{36}$ and $x_{43}$) are shown in the top plot. It can be seen that the system is steered from the initial cancerous attractor POA6 into the trajectories of the cyclic apoptosis attractor CYA, by implementing the control profiles of $u_{19}$ (average regulatory interaction strength from the other components to PI3 K/Akt) and $u_{41}$ (the average regulatory interaction strength from

the other components to TGF-β) as shown in the bottom plot. Using five state variables ($x_1$, $x_4$, $x_7$, $x_{18}$ and $x_{36}$), the top plot of figure 9b shows the realization of the state transition from the cancer attractor (POA6) to the normal gastric epithelial attractor (POA1). This is achieved by manipulating only one control variable $u_{19}$ and its profile is shown in the bottom plot of figure 9b. It is noted that $u_{19}$ is identified as a control variable in both scenarios, which is consistent with the result that gastric cancer can be transformed to the apoptosis attractor and the normal gastric epithelial attractor by consistently inhibiting the PI3 K/Akt activity [52].

## 4. Conclusion

Abnormal periodic rhythms lead to disorders of functionality of biological systems. In this study, we present an optimal control approach to reconciliating periodic rhythms to mitigate the effects of disorders. For this purpose, it is necessary to define an adequate intervention strategy. In this study, we consider this task as a state transition problem which is addressed by a mixed integer nonlinear dynamic programming approach. Our approach allows simultaneously identifying control variables and determining their profiles for intervention. In addition, our approach is able to determine the minimum number of control variables. Moreover, the control variables for intervention are constrained in allowable regions in the problem formulation, which can avoid excessive control actions and maintain the stability of biological systems. These features are of importance in the sense of clinical practice.

The results of three examples demonstrate the applicability of our approach. From a biological point of view, our optimal control approach explores the underlying mechanisms of state transition and phase resetting for biological networks on the system level. For instance, we reveal the key regulatory parameters to genes *PER2/CRY1* and *CLK/BMAL1* in mammalian circadian rhythms, which play an important role in regulating the dynamics when the system undergoes lower amplitude or a phase delay of circadian rhythms due to diet or time lag, respectively. Our results show that it is sufficient to nudge the undesired attractor into the target one by imposing intervention on the key regulatory parameters. In addition, the proposed approach can drive the gastric cancer attractor into the normal gastric epithelial attractor or apoptosis attractor for the gastric cancer gene regulatory network. This provides hints for alleviating gastric cancer. On the other hand, life processes are inherently stochastic, in which noises play an important role in biological systems from the aspect of single cell or at the molecular level [53]. Some studies have found that noises can help inducing the state transition [23]. Thus, consideration of noises and disturbances will be an important aspect in our future work. All in all, our work can be extended based on the studies on network description and analysis at the system level for understanding drug actions [54,55]. The application of the results of these studies will make our work a promising approach to improving the efficiency of drug design.

Data accessibility. The code and materials supporting our paper are provided as electronic supplementary material.
Authors' contribution. M.Y. and J.Q. conducted the numerical experiments; M.Y. performed the analyses and wrote the original manuscript; W.H. and P.L. discussed the results and contributed to the final manuscript. All authors read and approved the final manuscript.
Competing interests. The authors declare that they have no competing interests.
Funding. M.Y. was supported by the China Scholarship Council for 2-year study at Technische Universität Ilmenau (award no. 201706320236).

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
