## [Reviewer comments · Royal Society Open Science]

Review History

RSOS-191698.R0 (Original submission)

Review form: Reviewer 1

Is the manuscript scientifically sound in its present form?

Yes

Are the interpretations and conclusions justified by the results?

Yes

Is the language acceptable?

Yes

Do you have any ethical concerns with this paper?

No

Have you any concerns about statistical analyses in this paper?

No

Recommendation?

Accept as is

Comments to the Author(s)

I previously reviewed this manuscript as Reviewer 1 and thus I reviewed the changes and responses to my comments. I believe the authors adequately addressed each one of my points.

The new flow chart does make presentation clearer and the inclusion of the simulation files in the supplementary material makes the technical portions appropriately accessible to other researchers. The authors also performed additional simulations showing the robustness of their scheme.

I believe the work is scientifically sound and that the article is appropriate for *Royal Society Open Science*.

Minor typos:

1. Page 16 line 2 "sense of clinic practice." should be "clinical practice."

2. Figure 7 caption reads "due to a shift work." I think it is best as "due to shift work", as "shift work" is a somewhat technical term referring to work outside normal daytime hours. Otherwise it should be "a shift in work" or "a shift in work schedule".

Review form: Reviewer 2

Is the manuscript scientifically sound in its present form?

Yes

Are the interpretations and conclusions justified by the results?

Yes

Is the language acceptable?

Yes

Do you have any ethical concerns with this paper?

No

Have you any concerns about statistical analyses in this paper?

No

Recommendation?

Accept as is

Comments to the Author(s)

The authors have replied and or fixed the problems one by one. Though I do not think the contribution is enough for a publication in this journal, I do not give a direct conclusion of "reject". Alternatively, I will agree with the evaluations on contribution from the editor and other referees. The present version can be accepted as it be.

Decision letter (RSOS-191698.R0)

15-Nov-2019

Dear Professor Yuan,

On behalf of the Editors, I am pleased to inform you that your Manuscript RSOS-191698 entitled "Reconciling periodic rhythms of large-scale biological networks by optimal control" has been accepted for publication in Royal Society Open Science subject to minor revision in accordance with the referee suggestions. Please find the referees' comments at the end of this email.

The reviewers and handling editors have recommended publication, but also suggest some minor revisions to your manuscript. Therefore, I invite you to respond to the comments and revise your manuscript.

- Ethics statement

- Data accessibility

<http://datadryad.org/submit?journalID=RSOS&manu=RSOS-191698>

- Competing interests

- Authors' contributions

AB carried out the molecular lab work, participated in data analysis, carried out sequence alignments, participated in the design of the study and drafted the manuscript; CD carried out

the statistical analyses; EF collected field data; GH conceived of the study, designed the study, coordinated the study and helped draft the manuscript. All authors gave final approval for publication.

- Acknowledgements

- Funding statement

Because the schedule for publication is very tight, it is a condition of publication that you submit the revised version of your manuscript before 24-Nov-2019. Please note that the revision deadline will expire at 00.00am on this date. If you do not think you will be able to meet this date please let me know immediately.

- 1) A text file of the manuscript (tex, txt, rtf, docx or doc), references, tables (including captions) and figure captions. Do not upload a PDF as your "Main Document";
- 2) A separate electronic file of each figure (EPS or print-quality PDF preferred (either format should be produced directly from original creation package), or original software format);
- 3) Included a 100 word media summary of your paper when requested at submission. Please ensure you have entered correct contact details (email, institution and telephone) in your user account;
- 4) Included the raw data to support the claims made in your paper. You can either include your data as electronic supplementary material or upload to a repository and include the relevant doi within your manuscript. Make sure it is clear in your data accessibility statement how the data can be accessed;

5) All supplementary materials accompanying an accepted article will be treated as in their final form. Note that the Royal Society will neither edit nor typeset supplementary material and it will be hosted as provided. Please ensure that the supplementary material includes the paper details where possible (authors, article title, journal name).

If your manuscript is newly submitted and subsequently accepted for publication, you will be asked to pay the article processing charge, unless you request a waiver and this is approved by Royal Society Publishing. You can find out more about the charges at <https://royalsocietypublishing.org/rsos/charges>. Should you have any queries, please contact openscience@royalsociety.org.

Kind regards,
Lianne Parkhouse
Editorial Coordinator
Royal Society Open Science
openscience@royalsociety.org

on behalf of Dr Francois Fages (Associate Editor) and Marta Kwiatkowska (Subject Editor)
openscience@royalsociety.org

Associate Editor Comments to Author (Dr Francois Fages):

Dear authors,

Your revised paper has been reviewed by the previous reviewers who found it appropriate for publication as is.
Congratulations

Reviewer comments to Author:

Reviewer: 1
Comments to the Author(s)

I previously reviewed this manuscript as Reviewer 1 and thus I reviewed the changes and responses to my comments. I believe the authors adequately addressed each one of my points.

The new flow chart does make presentation clearer and the inclusion of the simulation files in the supplementary material makes the technical portions appropriately accessible to other researchers. The authors also performed additional simulations showing the robustness of their scheme.

I believe the work is scientifically sound and that the article is appropriate for *Royal Society Open Science*.

Minor typos:

1. Page 16 line 2 "sense of clinic practice." should be "clinical practice."
2. Figure 7 caption reads "due to a shift work." I think it is best as "due to shift work", as "shift work" is a somewhat technical term referring to work outside normal daytime hours. Otherwise it should be "a shift in work" or "a shift in work schedule".

Reviewer: 2

Comments to the Author(s)

The authors have replied and or fixed the problems one by one. Though I do not think the contribution is enough for a publication in this journal, I do not give a direct conclusion of "reject". Alternatively, I will agree with the evaluations on contribution from the editor and other referees. The present version can be accepted as it be.

Author's Response to Decision Letter for (RSOS-191698.R0)

See Appendix A.

Decision letter (RSOS-191698.R1)

28-Nov-2019

Dear Professor Yuan,

It is a pleasure to accept your manuscript entitled "Reconciling periodic rhythms of large-scale biological networks by optimal control" in its current form for publication in *Royal Society Open Science*. The comments of the reviewer(s) who reviewed your manuscript are included at the foot of this letter.

Please note that the email address "Junlin.qu@tu-ilmenau.de" is not accepting our emails - you must supply an alternative email address for Dr Qu.

Please ensure that you send to the editorial office an editable version of your accepted manuscript, and individual files for each figure and table included in your manuscript. You can send these in a zip folder if more convenient. Failure to provide these files may delay the

processing of your proof. You may disregard this request if you have already provided these files to the editorial office.

on behalf of Dr Francois Fages (Associate Editor) and Marta Kwiatkowska (Subject Editor)
openscience@royalsociety.org

Appendix A

Dear Editor,

Thank you for your recognition of our research work presented in this paper. We have revised our manuscript in accordance with the referee suggestions. The revised parts are marked in red in the version of "marked Main Document". Also, we have uploaded the 'clean' version of the new manuscript as "clean Main Document". Our responses to the reviewers' comments are as follows:

Reviewer #1:

1. Comment: I previously reviewed this manuscript as Reviewer 1 and thus I reviewed the changes and responses to my comments. I believe the authors adequately addressed each one of my points. The new flow chart does make presentation clearer and the inclusion of the simulation files in the supplementary material makes the technical portions appropriately accessible to other researchers. The authors also performed additional simulations showing the robustness of their scheme. I believe the work is scientifically sound and that the article is appropriate for *Royal Society Open Science*.

Response: *Thanks for this comment.*

2. Comment: Page 16 line 2 "sense of clinic practice." should be "clinical practice".

Response: *We have revised 'clinic' to 'clinical'.*

3. Comment: Figure 7 caption reads "due to a shift work." I think it is best as "due to shift work", as "shift work" is a somewhat technical term referring to work outside normal daytime hours. Otherwise it should be "a shift in work" or "a shift in work schedule".

Response: *Thanks for pointing this out. We have revised it into 'due to shift work'.*

Reviewer #2:

1. Comment: The authors have replied and or fixed the problems one by one. Though I do not think the contribution is enough for a publication in this journal, I do not give a direct conclusion of "reject". Alternatively, I will agree with the evaluations on contribution from the editor and other referees. The present version can be accepted as it be.

Response: *Thanks.*

Yours sincerely,

Meichen Yuan, Junlin Qu, Weirong Hong, Pu Li